# Chemical Properties of a Volcanic Soil Are Influenced by Eight Years of Crop Rotations with Different Levels of Residue Incorporation

**DOI:** 10.3390/plants14050764

**Published:** 2025-03-02

**Authors:** Juan Hirzel, Pablo Undurraga, Carola Vera, Iván Matus, Mauricio Schoebitz

**Affiliations:** 1Instituto de Investigaciones Agropecuarias, INIA Quilamapu, Chillán 3800062, Chile; pundurra@inia.cl (P.U.); carola.vera@inia.cl (C.V.); imatus@inia.cl (I.M.); 2Department of Soil and Natural Resources, Faculty of Agronomy, Universidad de Concepción, Concepción 4030000, Chile; mschoebitz@udec.cl

**Keywords:** conservation agriculture, crop rotations, residue recycling, soil fertility, volcanic soil

## Abstract

The objective of this study was to evaluate the effect of four cycles of six biannual rotations and four levels of incorporation of residues for each crop on the chemical properties of a volcanic soil from central-southern Chile. Methods: After six biannual rotations (canola–bread wheat, bean–bread wheat, canola–durum wheat, bean–durum wheat, canola–corn, and bean–corn) and four levels of residue incorporation (0, 50, 100, and 200%), we evaluated the chemical properties of a volcanic soil through eight years of cultivation. Results: The chemical properties of the soil were affected mainly by crop rotation and to less extent by the dose of residue incorporated. Beans showed a positive relation with soil pH, unlike canola with a negative effect (*p* < 0.05). Corn was also noticeably negative for available P. The application of increasing doses of residue positively affected the exchangeable K and Mg (*p* < 0.01). There were also positive correlations between pH and exchangeable Ca, exchangeable Ca and Mg, and available S and exchangeable Al and negative correlations between pH and exchangeable Al, pH and available S, and available S with exchangeable Ca and Mg. Conclusions: Carrying out different crop rotations seems to be a boost for chemical properties of the soil, while the incorporation of residues allows higher concentrations of exchangeable K and Mg.

## 1. Introduction

Crop rotation generates a beneficial effect over yield and quality [1] and can even reduce dependence on the use of herbicides due to the suppressive effect that crops like canola (*Brassica napus* L.) have on weeds [2]. There are additional benefits on crop production such as nitrogen (N) and water use efficiency [3]. With the inclusion of legumes, beneficial effects on the chemical and biological properties of the soil positively affect the production of the following harvest [1,4,5,6,7]. Moreover, rotations can generate time windows between harvests, allowing the incorporation of residues, bringing short term benefits for soil fertility and environment [6,8,9].

The global production of crop residues has been growing, associated with the increase in demand for grain foods and their production, registering a volume of 4 billion tons annually [10,11]. Although this waste production can generate carbon and nutrient recycling when used in agriculture [12], its inadequate management or its disposal through practices such as burning negatively affects the environment [8,10].

In general, the nutritional composition of non-legume crop residues presents carbon (C) concentrations between 40–45% (40–50% cellulose, 15–25% hemicellulose, and 20–30% lignin), N between 0.60–1.12%, phosphorus (P) between 0.045–0.197%, and potassium (K) between 1.26–2.35% [8]. When those nutrients enter the soil, they feed the turnover of C and nutrients, with benefits in biological, physical, and chemical properties and in plant production [8,13].

Mutai et al. [14] report that the incorporation of corn residues for 45 consecutive years in the first 20 cm of a clay loam soil in Kenya increased organic matter content, pH, and stability of aggregates compared to the control without residues. In turn, for all treatments in this study (application of cattle manure, fertilizers, and residues), there was a positive and significant correlation between the organic matter content and available phosphorus.

Similarly, Hou et al. [10] indicated that the incorporation of sweet corn residues mixed with chemical fertilizers as a fertilization source for three consecutive seasons led to an increase in soil pH and the contents of organic matter and total nitrogen.

Xia et al. [15], in a meta-analysis, describe how incorporation of residues showed increases in the available fractions of N, P, and K of 14, 10, and 18%.

Diop et al. [4] mention that, with respect to the control, the incorporation of residues in a farmless rotation system improved the cation exchange capacity and available sulfur at a depth of 10 to 20 cm of a clay soil, with no effect on other chemical properties in both strata analyzed (0–10 and 10–20 cm).

In a study conducted in a sandy soil from Russia, Mukhametov et al. [7] indicated that the incorporation of residues in crop rotations that included the legumes velvet beans (*Mucuna pruriens* L.), soybeans (*Glycine max* L.), and vetch (*Vicia sativa* L.) prior to corn raised the pH and available P of the soil, without significant effect on organic C, available N, Ca, Mg, and exchangeable K.

Data are also available from Bangladesh, in a silty loam soil where the increase from 15 to 40% in the dose of the residue incorporated from a lentil (*Lens culinaris* L.)–mungbean (*Vigna radiata* L.)–rice (*Oriza sativa* L.) rotation was followed by more organic and inorganic fractions of P in the first 30 cm, in addition to raising soil organic matter [16].

In general, farmers are unaware of the benefits of incorporating waste, avoiding this practice due to operational difficulties, as well as the risks of partial alteration in the cycle of some nutrients with a high impact on the growth rate, such as N. Hou et al. [10] highlighting that the C:N ratio in wheat (*Triticum aestivum* L.) and corn (*Zea mays* L.) residues correspond to 80–100:1 and 50–70:1, respectively, which initially generates N immobilization in the soil, taking in consideration that the C:N ratio that facilitates mineralization is less than 25 [8]. However, field studies in different soil conditions, residue levels, and incorporation techniques have indicated a beneficial effect in the medium term on the productivity of different indicator crops [1,10,17].

The application of crop rotations with incorporation of residues is a conservationist practice that should improve in Latin America. We theorize that the incorporation of residues in increasing doses in crop rotations gives a time window between harvesting and sowing, with consequent improvement in soils’ chemical properties. The objective of this study was to evaluate the effect of four cycles of six biannual rotations (canola–bread wheat, bean–bread wheat, canola–durum wheat, canola–durum wheat, canola–corn, and bean–corn) and four levels of incorporation of residues for each crop (0%, 50%, 100%, and 200%) on the chemical properties of a volcanic soil from central-southern Chile at the end of the evaluation period.

## 2. Materials and Methods

### 2.1. Site Characteristics

The experiment was conducted for eight consecutive seasons from 2016 to 2024 at the Santa Rosa Experimental Station, INIA-Quilamapu, Chillán, Chile (36°31′ S; 71°54′ W), in a volcanic soil (Melanoxerand) [18] with a loamy texture, acidic soil reaction, low bulk density, and a high organic matter content (Table 1). The climate is temperate Mediterranean, characterized by a hot, dry summer and cold, wet winter, and its annual precipitation, temperature, and evaporation are descripted in Table 2.

After four cycles of six biannual crop rotations and four levels of residue incorporation in each crop (0, 50, 100, and 200%), measurements and soil chemical properties analysis were performed at the end of the eighth season (May 2024).

### 2.2. Crops Management

The management and details of each crop are described in Hirzel et al. [19]. The experiment consisted of biannual rotations of six crop combinations: canola–bread wheat, canola–durum wheat (*Triticum durum* L.), canola–corn, bean (*Phaseolus vulgaris* L.)-bread wheat, bean–durum wheat, and bean–corn, in which residues of the previous crop were incorporated at levels of 0%, 50%, 100%, and 200%. The experimental unit for each crop rotation was 40 m long and 14 m wide (560 m^2^), which was divided in four sub-plots (140 m^2^) each for residue levels 0, 50, 100, and 200%. The treatment with the 200% residue level was achieved by retaining all the residues from previous crops plus residue imported from the control (0%). Considering six crop rotations and four replicates for each rotation, the total experimental area was 13,440 m^2^, with four levels of residue incorporation for each crop rotation. This design was maintained throughout the 8 seasons. At the start of the biannual rotation experiment (year 2016), lime was applied at a rate of 3000 kg·ha^−1^ prior to sowing of all the crops (1.34 ha) in April 2016 to correct soil acidity (Table 1).

### 2.3. Soil Analysis

At the beginning and end of the eighth season, composite samples were manually collected from the topsoil (0–20 cm) for each treatment on the same day corn was harvested. All samples were air-dried and sieved with a 2 mm mesh. Chemical protocols are as described by Sadzawka’s methods [20].

### 2.4. Experimental Design and Statistical Analysis

The experimental design was split plot in completely randomized blocks, where the main plots were the six crop rotations and the sub-plots were the residue levels (four levels) with four replicates. Results were analyzed by ANOVA and Tukey’s test (*p* = 0.05) using the SAS PROC MIXED Model procedure 6.0 (SAS Institute, Cary, NC, USA) [21]. For significant interactions, contrast analysis was used to compare the treatment effects separately. In addition, Pearson’s multiple correlation analysis was performed to assess the relationship between soil chemical properties.

## 3. Results

### 3.1. Edaphoclimatic Conditions of Experimental Site and Production

Soil analyses at the beginning of the experiment only indicated acidity limitations, which were corrected with soil liming (Table 1). The climatic conditions during the 8 years of crop rotation were adequate for the cultivated species, except for the 2023–2024 season (Table 2), in which there was an increase in precipitation reaching the month of October, which delayed the sowing of the bread wheat, durum wheat, and corn crops, mainly affecting the yield of the bread wheat crop (Table 3), presenting a yield loss of 24 and 25% after canola and beans compared to the average obtained in previous cycles. For durum wheat, there was a 19 and 14% yield loss after canola and bean compared to the average obtained in previous cycles. For corn, the yield loss after canola and bean corresponded to 0 and 9% compared to the average obtained in the previous cycles.

### 3.2. Significance Analysis of Soil Chemical Parameters

All soil chemical parameters analyzed were affected by crop rotation (*p* < 0.05), while the applied residue dose only had an effect on the concentrations of exchangeable Mg and K (*p* < 0.05) (Table 4). There was an interaction effect on the concentration of available N, concentration of NH_4_^+^-N, and exchangeable Na (Table 4); therefore, these interactions were analyzed in each rotation separately.

### 3.3. Effect of Rotations and Residue Doses on Soil Chemical Properties

The chemical properties of the soil evaluated at the end of the eighth year of crop rotation (four cycles of biennial rotations) (Table 5) indicate that the soil pH was lower in the rotations that included canola (*p* < 0.05) and higher in two of the rotations that included beans. Soil OM was higher in two of the rotations that included canola (*p* < 0.05), and the lowest value came from rotations that included durum wheat (*p* < 0.05). Rotations that included corn achieved intermediate to high values, highlighting the canola–corn combination (Table 5). For available N and nitric N, the highest value was achieved in the canola–bread wheat and bean–durum wheat rotations (*p* < 0.05) and the lowest values in canola– and bean–corn rotations. Ammonium N presented a behavior similar to available N, with a higher value in the canola–bread wheat and bean–durum wheat rotations (*p* < 0.05). Available P was higher in the two bread wheat rotations (*p* < 0.05) and lower in the bean–corn rotation.

The exchangeable Ca presented the lowest value in the canola–durum wheat rotation, with statistical similarity to the value obtained with the bean–corn rotation (*p* < 0.05). For the exchangeable Mg, the highest values were achieved in the rotations that combined bean with both types of wheat (*p* < 0.05), and the lowest value in the canola–durum wheat rotation (Table 5). The highest value of K was obtained in the canola–corn rotation (*p* < 0.05), with no differences between others. Na had the highest values in both rotations that included corn, but they only significantly exceeded rotations including durum wheat, highlighting the lower value obtained in the canola–durum wheat rotation (*p* < 0.05). Regarding exchangeable Al, the canola–corn rotation was the highest, similar to the value obtained with the canola–bread wheat rotation (*p* < 0.05). And for available S, two of the rotations that included canola had better values and bean gave the lowest (*p* < 0.05).

We report that the concentrations of both exchangeable Mg and K presented a relationship directly proportional to the dose of residue entered, with an effect of increasing concentration from the incorporation of 100% of the residue (*p* < 0.05) (Table 6).

### 3.4. Effect of Crop Rotation and Residue Dose Interactions on Soil Chemical Properties

The available N presented an interaction only in the canola–bread wheat rotation (Figure 1), and the highest dose of residue allowed the concentration of this nutrient to increase, with no difference between the other doses of residue (*p* < 0.05).

Ammoniacal N showed interaction in three of the six rotations evaluated (Figure 2, Figure 3 and Figure 4 ). In the canola–bread wheat rotation (Figure 2) the highest values were achieved with the three doses of residue incorporated without differences between them (*p* < 0.05). In contrast, in the bean–bread wheat rotation (Figure 3), the highest value was achieved in the control without residue and the lowest values in the three treatments with residue application (*p* < 0.05). For its part, in the canola–durum wheat rotation (Figure 4), the highest value was achieved with the highest dose of residue (*p* < 0.05), with no differences between the other treatments (*p* > 0.05).

For the exchangeable Na, there was interaction in two of the rotations evaluated. In the canola–bread wheat rotation (Figure 5), the highest values were obtained in the control and in the lowest dose of residue (*p* < 0.05), with no differences in the treatments with higher doses of residue (*p* > 0.05), while in the bean–corn rotation (Figure 6) there were erratic effects of the residue doses, with a higher value when 100% of the residue dose was applied, and the lowest values with the doses of 50 and 200% residue (*p* < 0.05).

### 3.5. Correlation Analysis of Soil Chemical Properties

Although there were several significant correlations between the soil properties analyzed (Table 7), only some are important due to their moderate to high values (≥0.39), such as the inversely proportional relationships pH–OM, pH–available P, pH–exchangeable Al, pH–available S, exchangeable Ca–Al, exchangeable Mg–Al, exchangeable Ca–available S, exchangeable Mg–Al, and exchangeable Mg–available S, and also the directly proportional relationships pH–exchangeable Ca, pH–exchangeable Mg, OM–exchangeable Al, exchangeable Ca–Mg, exchangeable Ca–K, exchangeable Mg–K, and exchangeable Al–available S. In turn, high correlation and significance values were obtained between available N, ammoniacal N, and nitrate N (Table 7).

## 4. Discussion

The yields obtained in the study area are within the potential expected of each crop [6,22,23,24], except for the eighth year of rotation, where the bread wheat yield was affected by late sowing, accelerating its phenological stages and the production-remobilization of carbohydrates to the grain. Due to abundant spring rains, planting of durum wheat and corn was late in that season; nevertheless, genotypes had good productive performance, moderately affecting durum wheat and slightly affecting corn.

The effect obtained by crop rotation on the chemical properties of the soil, as an average of the four levels of incorporated residue, was expected, mainly in crops with a high extraction capacity of mineral nutrients, such as corn and canola [13,22], and crops that stimulate greater biological activity in the soil, such as beans [5,6,7,16].

The greatest decrease in soil pH obtained in the rotations that included canola could be explained by the high extraction of basic reaction nutrients (Ca and Mg mainly), causing physiological acidity [13,22]. In contrast, a higher pH was achieved in the rotations that included bean, associated with its lower nutrient extraction [13,22]. Canola cultivation releases sulfur compounds into the soil [25], which increases the generation of protons and thus acidification [13,26].

When including high-carbon-generation crops like canola, higher values of organic matter were registered. The same effect would have been expected in rotations that included corn, which was somewhat obtained in our experiment in the canola–corn rotation. An explanation may be the decomposition speed of each residue, associated with its C/N ratio, which is lower in canola and higher in corn [8,10,27].

Available N, as an average of the four residue levels, was lower than the initial value of the experiment, which can be explained by the introduction of residues with a C/N ratio greater than 25, except for beans, which decreases the concentration of available N [8,13]. A higher value of available N was expected to be found in rotations that included beans [7,28], however, in only one of these rotations (bean–durum wheat) was saw this effect. In contrast, in the bean–corn rotation a lower concentration of available N was achieved, despite the high dose of N used in the corn crop. This might be due to a higher dose of residue incorporated with the corn crop, as well as its high C/N ratio compared to other crops such as wheat [10]. On the other hand, the dose of N used in the durum wheat crop (240 kg ha^−1^) is greater than its consumption (yields from 5.01 to 7.40 Mg ha^−1^, Table 3, and a consumption of 30 kg of N in the crop to achieve 1 Mg of grain) [22], which allows the generation of a higher level of available N in the soil when this crop is included in the rotation, and even higher when it is rotated with a legume.

Concentrations of ammoniacal and nitrate N are similar to available N; however, in all rotations, the concentration of nitric N was quantitatively higher than that of ammoniacal N, which should happen in soils with a high content of organic matter that favor biological processes, like nitrification [13,28].

Regarding P, and as an average of four levels of residue, the highest concentrations being found in the rotations that included bread wheat can be explained by the use of P doses (120 kg of P_2_O_5_ ha^−1^) much higher than its consumption (average of 26.8% of P extracted with respect to the applied dose), given that this crop presents a consumption of 2.55 kg of P in the crop to produce 1 Mg of grain [22], with yields that fluctuated between 4.52 to 6.57 Mg ha^−1^ (Table 3). Similarly, the lowest value of available P being obtained in the bean–corn rotation is related to the balance of P inputs and outputs in the soil during the 8 years of rotation, considering a dose of 120 kg of P_2_O_5_ ha^−1^ in the corn crop and 60 kg of P_2_O_5_ ha^−1^ in the bean crop, with extractions of 2.78 kg of P in the corn crop to produce 1 Mg of grain, and 1.8 kg of P in the bean crop to produce 1 Mg of grain [22]. Yields were from 14.22 to 16.27 Mg ha^−1^ in the corn crop and from 3.30 to 4.71 Mg ha^−1^ in the bean crop (Table 3).

The bean–corn rotation resulted in a higher pH, which could also have affected the availability of P in the soil, due to a probable greater formation of Ca–P and Mg–P complexes [13]. On the other hand, the high percentage of phosphorus fixation in amorphous clays and in Fe and Al oxides and hydroxides present in this volcanic soil must be considered [13,26,29].

Exchangeable Ca values were very stable between rotations, and the lowest value, obtained in the canola–durum wheat rotation, is associated with the lower pH obtained in these rotations and the replacement of Ca^2+^ by H^+^ in the cation exchange sites [13,26]. However, in two of the other rotations in which a lower pH was obtained (canola–bread wheat and canola–corn) and similar to that obtained in the canola–durum wheat rotation, this effect on exchangeable Ca was not seen. This is probably due to the greater extraction of bases generated in the durum wheat crop compared to bread wheat, considering its higher accumulated production during eight years, and the recycling of bases with the corn crop, associated with its better production of residues. In general, the concentration of exchangeable Ca in all rotations, except in canola–durum wheat, was higher than the value obtained at the beginning of the rotations, which was expected given the practice of applying CaCO_3_ × MgCO_3_ at doses of 3 Mg ha^−1^ carried out at the beginning of the experiment.

For exchangeable Mg, we did not find a relation to pH [26]. However, a higher value of exchangeable Mg was obtained in those rotations with lower grain production (bean–bread wheat and bean–durum wheat) as a result of a lower extraction of this nutrient from the soil [22,27]. In all crops, the Mg dose used corresponded to 10.78 kg ha^−1^ year^−1^, which is lower than the amount extracted with the grain harvest by canola and corn crops, similar to the extraction with the grain harvest in both wheat crops, and slightly higher than the amount extracted in the grains with the bean crop [22,27]. This generates Mg balances in the soil that are positive for the bean crop, neutral for both wheat crops, and negative for canola and corn, which may explain the lower values of exchangeable Mg obtained in the rotations that included canola and corn.

The K doses used in the crops of each rotation were generally higher than the extractions carried out with the grain harvest, so a higher concentration of exchangeable K than that obtained at the beginning of the experiment would have been expected; however, the practice of applying CaCO_3_ × MgCO_3_ generated cation exchange processes that could have resulted in K retrogradation or movement of this nutrient to a greater depth [13]. The higher value of exchangeable K obtained in the canola–corn rotation, as an average of the four residue levels, can be explained by the greater mass of residue incorporated in this rotation, which allows the return of between 70 and 80% of the extracted K [22,27]. The higher concentration of exchangeable Na obtained in the rotations that included corn can be explained by the greater mass of residues incorporated into the soil, allowing the recycling of the extracted Na, as well as by a greater extraction of Ca and K, which allow the release of cation exchange sites to be occupied by other positively charged nutrients such as Na [13,26].

The exchangeable Al showed an expected behavior based on the pH obtained in each rotation, with higher values in two of the rotations with lower pH. In general, exchangeable Al diminished after the first values registered.

We anticipated higher value of available S, given the high concentrations of S in the canola plant and incorporation with recycling residues [13,22,27].

With respect to the residue doses used, and as an average of the crop rotations, the organic matter in soil should increase in proportion to the mass of carbon incorporated [10,14]; nevertheless, the volcanic soil has a very high content of organic matter, probably diluting entering carbon. In addition, once the carbon enters, it is oxidized by the soils biomass, generating a loss of 2/3 parts through respiration [13,28].

In relation to soil pH, an increase associated with the dose of residue and its contribution of basic reaction cations was likely to occur [10,14,27]; however, in volcanic soils with a high percentage of organic matter such as the one used in this experiment, the mineralization rates of C and N generate acidic reaction compounds that modulate an increase in pH [13,28]. As for available N, a decrease associated with the increasing dose of residue was predicted; however, in soils with a high percentage of organic matter, the mineralization of native organic N, as well as that introduced with the residues, could maintain stable concentrations of N in the soil [13,28]. With the increase in the residue dose, an increase in the concentration of available P should occur [15,16]; however, soils with amorphous clays such as the one used in this experiment have a high P fixation coefficient [13,26,29].

Among the main nutrients whose distribution in the plant is greater in the residue, Ca, Mg, and K stand out [13,22,27]; however, in this experiment, the increasing dose of residue had an effect only on the concentrations of exchangeable K and Mg, similar to that indicated by Xia et al. [15] for K. The absence of effect on exchangeable Ca may be explained by the intensity of the adsorption processes of this cation in the soil, associated with its smaller hydration radius with respect to other cations [13,30].

The interactions between crop rotation and residue dose had slight influence on soil chemical properties. A higher residue dose would have been expected to decrease the concentration of available N, ammonium N, or nitrate N in some rotations; however, this was only observed for ammonium N in the bean–bread wheat rotation, where the residue mass from durum wheat was greater than the contribution generated by the bean, which generated an accumulated C/N ratio over the 8 years of rotation greater than 25 [8]. However, the quantitative differences in the concentration of ammonium N are very low (less than 4 mg kg^−1^). An unexpected effect was seen in the canola–bread wheat rotation, with an increase in the concentrations of available N and ammoniacal N when the highest dose of residue was used, especially considering that the C/N ratios of these crops were 41 and 76, respectively [27]. However, the concentration differences in quantitative terms were also low (5 and 2 mg kg^−1^ for available N and ammoniacal N). Similarly, this effect was observed for the concentration of ammoniacal N in the canola–durum wheat rotation, with a quantitative difference of 4 mg kg^−1^ when using the highest dose of residue compared to the other treatments. For exchangeable Na, there was a clear effect of decreasing concentration with the highest residue doses used in the canola–bread wheat rotation, which may respond to cation exchange processes generated with the input of Ca, Mg, and K in the residues, and to a greater magnitude with the 100 and 200% residue doses [13,27]. However, the difference in magnitude was very low (0.02 to 0.03 cmol_+_ kg^−1^). The differences in exchangeable Na in the bean–corn rotation were erratic and do not allow us to infer an effect of the residue dose.

The correlations obtained among the chemical properties were not surprising. The inversely proportional relationship between pH and exchangeable Al associated with the increase in proton concentration and loss of basic reaction cations from the exchange sites is documented, as well as pH and OM, since the mineralization of the latter generates direct or indirect production of protons and in turn an increase in nutrients present in the organic fraction such as available S [13,26]. Likewise, there is a directly proportional relationship between the increase in basic reaction cations in the soil such as Ca and Mg and its pH (increase in relative concentration of OH^-^ and decrease in concentration of H^+^) [13,26]. Soil acidification caused by OM mineralization leads to an increase in exchangeable Al as a replacement effect of basic reaction cations [13,26]. On the other hand, the decrease in H^+^ from cation exchange sites allows for an increase in basic reaction cations that were used for initial soil liming (Ca and Mg) and for annual fertilization of crops in each rotation (K and Mg), as well as those generated by recycling residues (Ca, Mg, and K) [13,26,27].

It was also expected that those elements directly related to the increase in pH (Ca and Mg) would present an inversely proportional relationship with available S, as observed in the pH–available S relationship. Likewise, the elements inversely proportional to pH, such as exchangeable Al and available S, presented a directly proportional relationship with each other. Finally, the high correlations found between available N and the ammoniacal and nitrate fractions were also as predicted, although the highest value was obtained between available N and nitrate N, which is normal in soils with high OM content and adequate oxygenation, since nitrification processes are favored [13,27].

## 5. Conclusions

The use of the six crop rotations with different levels of residue incorporation had an effect on the chemical properties of the soil mainly due to the rotation used and to a lesser extent due to the residue dose. None of the rotations improved all the chemical properties evaluated, highlighting only a positive effect of bean on soil pH in two of the three rotations in which this crop was included and the negative effect of canola when used. Also noteworthy is the negative effect of corn on available P. The application of increasing doses of residue positively affected exchangeable K and Mg. On the other hand, the correlations obtained in the chemical properties of the soil were mostly consistent with what has been reported in the literature, highlighting the high positive correlations between pH and exchangeable Ca, exchangeable Ca and Mg, and available S and exchangeable Al and the high negative correlations between pH and exchangeable Al, pH and available S, and available S and exchangeable Ca and Mg. We suggest that using different crop rotations is a tool to improve certain soil chemical properties, while incorporating residues allows higher concentrations of exchangeable K and Mg.

## Figures and Tables

**Figure 1 plants-14-00764-f001:**
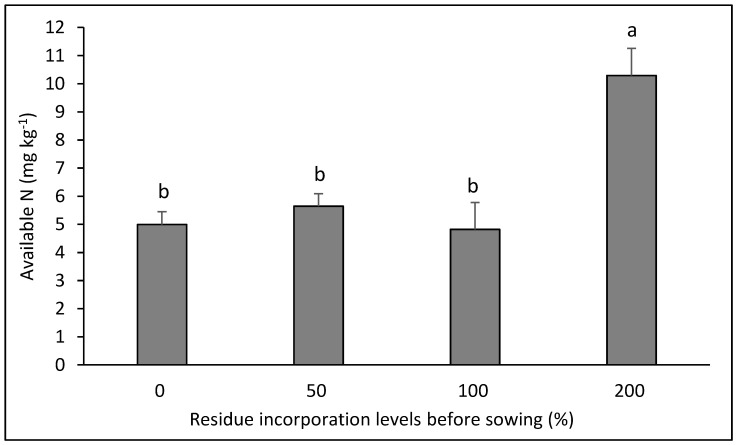
Soil available N after eight years in the crop rotation canola–bread wheat as affected by four levels of residue incorporation as an average of six crop rotations. Different letters above the bars indicate differences according to Tukey’s test (*p* < 0.05). Whiskers correspond to the standard error for each bar.

**Figure 2 plants-14-00764-f002:**
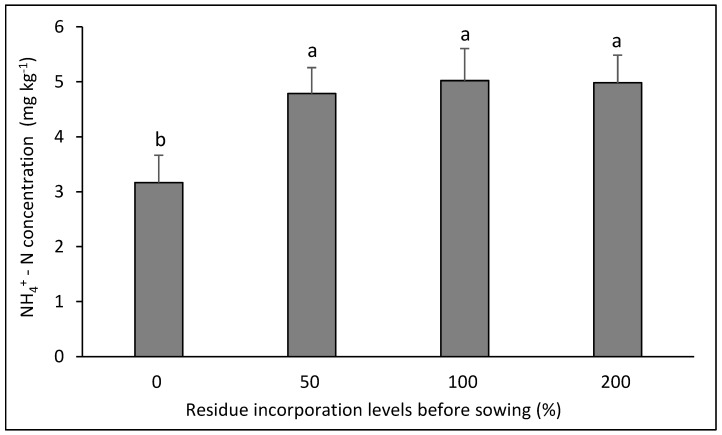
Soil NH_4_^+^–N concentration after eight years in the crop rotation canola–bread wheat as affected by four levels of residue incorporation as an average of six crop rotations. Different letters above the bars indicate differences according to Tukey’s test (*p* < 0.05). Whiskers correspond to the standard error for each bar.

**Figure 3 plants-14-00764-f003:**
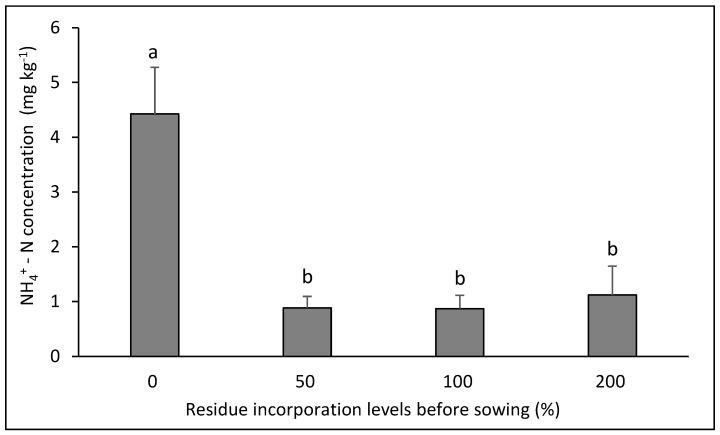
Soil NH_4_^+^–N concentration after eight years in the crop rotation bean–bread wheat as affected by four levels of residue incorporation as an average of six crop rotations. Different letters above the bars indicate differences according to Tukey’s test (*p* < 0.05). Whiskers correspond to the standard error for each bar.

**Figure 4 plants-14-00764-f004:**
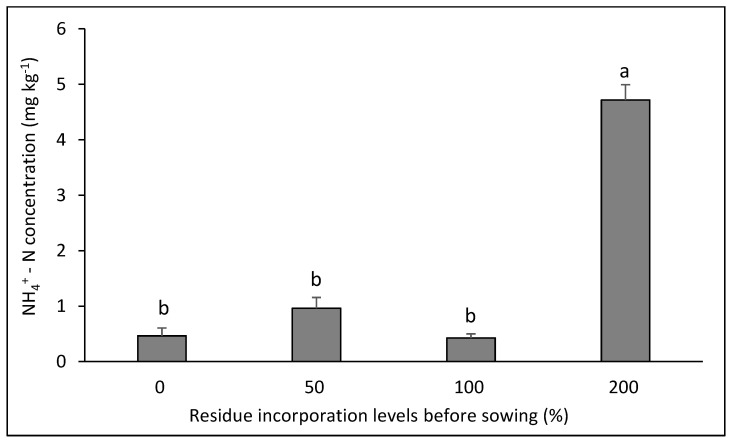
Soil NH_4_^+^–N concentration after eight years in the crop rotation canola–durum wheat as affected by four levels of residue incorporation as an average of six crop rotations. Different letters above the bars indicate differences according to Tukey’s test (*p* < 0.05). Whiskers correspond to the standard error for each bar.

**Figure 5 plants-14-00764-f005:**
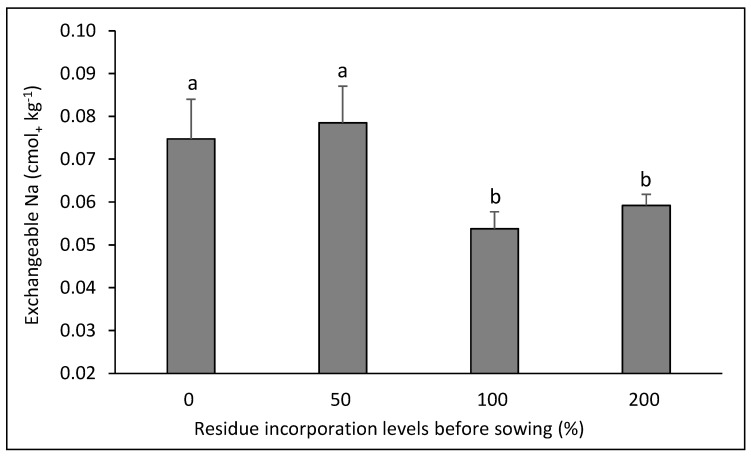
Soil exchangeable Na after eight years in the crop rotation canola–bread wheat as affected by four levels of residue incorporation as an average of six crop rotations. Different letters above the bars indicate differences according to Tukey’s test (*p* < 0.05). Whiskers correspond to the standard error for each bar.

**Figure 6 plants-14-00764-f006:**
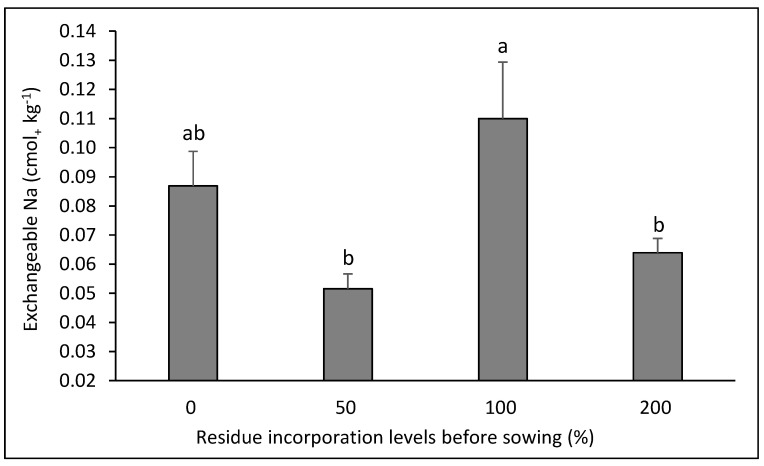
Soil exchangeable Na after eight years in the crop rotation bean–corn as affected by four levels of residue incorporation as an average of six crop rotations. Different letters above the bars indicate differences according to Tukey’s test (*p* < 0.05). Whiskers correspond to the standard error for each bar.

**Table 1 plants-14-00764-t001:** Pre-cropped soil chemical properties during the 2015–2016 experimental season.

Parameters	Value
Clay, %	16.7
Silt, %	44.6
Sand, %	38.7
Bulk density, g cm^−3^	1.00
pH _(soil:water 1:5)_	5.52
Organic matter, g kg^−1^	109.2
EC, dS m^−1^	0.11
Available N, mg kg^−1^	54.1
Available P, mg kg^−1^	21.3
Exchangeable K, cmol_+_ kg^−1^	0.54
Exchangeable Ca, cmol_+_ kg^−1^	4.20
Exchangeable Mg, cmol_+_ kg^−1^	0.36
Exchangeable Na, cmol_+_ kg^−1^	0.08
Exchangeable Al, cmol_+_ kg^−1^	0.12
Available S, mg kg^−1^	23.5

EC: electrical conductivity; N: nitrogen; P: phosphorus; K: potassium; Ca: calcium; Mg: magnesium; Na: sodium; Al: aluminum; S: sulfur.

**Table 2 plants-14-00764-t002:** Climatic characteristics of the experimental site during the first eight years of development of crops rotation.

Season	Medium Temperature (°C)	Precipitation (mm)	Evaporation (mm)
2016	12.8	605	1023
2017	13.2	563	1041
2018	13.5	730	990
2019	13.4	632	994
2020	13.6	746	1077
2021	13.5	649	940
2022	13.2	920	966
2023	13.7	1209	925

**Table 3 plants-14-00764-t003:** Production of grain (Mg ha^−1^) during the first eight years of crop rotations as average of four levels of residue incorporation at the end of each crop.

Crop Rotation	2016–2017		2017–2018		2018–2019		2019–2020		2020–2021		2021–2022		2022–2023		2023–2024	
C–BW	C	4.17	C	--	C	4.31	C	--	C	2.92	C	--	C	2.65	C	--
	BW	--	BW	5.74	BW	--	BW	5.77	BW	--	BW	6.57	BW	--	BW	4.54
B–BW	B	3.75	B	--	B	3.67	B	--	B	3.98	B	--	B	3.45	B	--
	BW	--	BW	6.15	BW	--	BW	5.96	BW	--	BW	6.08	BW	--	BW	4.52
C–DW	C	4.23	C	--	C	4.35	C	--	C	2.96	C	--	C	2.80	C	--
	DW	--	DW	6.05	DW	--	DW	6.02	DW	--	DW	6.47	DW	--	DW	5.01
B–DW	B	4.12	B	--	B	4.46	B	--	B	4.41	B	--	B	3.93	B	--
	DW	--	DW	7.38	DW	--	DW	7.40	DW	--	DW	6.44	DW	--	DW	6.10
C–Corn	C	4.45	C	--	C	4.56	C	--	C	3.48	C	--	C	3.15	C	--
	Corn	--	Corn	16.86	Corn	--	Corn	15.80	Corn	--	Corn	15.21	Corn	--	Corn	16.06
B–Corn	B	4.02	B	--	B	4.42	B	--	B	4.71	B	--	B	3.30	B	--
	Corn	--	Corn	16.27	Corn	--	Corn	15.58	Corn	--	Corn	14.89	Corn	--	Corn	14.22

C: canola; BW: bread wheat; DW: durum wheat; B: bean.

**Table 4 plants-14-00764-t004:** Test of significance of the soil chemical properties as affected by six crop rotations sequences with four levels of residue incorporation.

Soil Properties	Crop Rotation (CR)	Residue Level (R)	CR × R Interaction
pH	0.0001	0.39	0.75
Organic matter	0.0001	0.17	0.053
Available N	0.0001	0.83	0.015
NH_4_^+^–N	0.0001	0.30	0.0001
NO_3_^−^–N	0.0001	0.55	0.42
Available P	0.0001	0.44	0.08
Exchangeable Ca	0.0016	0.25	0.24
Exchangeable Mg	0.0001	0.0026	0.39
Exchangeable K	0.0003	0.0001	0.69
Exchangeable Na	0.0001	0.053	0.0003
Exchangeable Al	0.0001	0.20	0.60
Available S	0.0001	0.49	0.88

**Table 5 plants-14-00764-t005:** Soil chemical properties as affected by six crop rotations with four levels of residue incorporation.

Soil Properties	Crop Rotation					
	C–BW	B–BW	C–DW	B–DW	C–Corn	B–Corn
pH	5.97 c	6.04 bc	5.99 c	6.11 ab	5.99 c	6.16 a
OM, g kg^−1^	115.0 a	106.0 bc	96.0 d	92.0 d	110.0 ab	103.0 c
Available N, mg kg^−1^	18.3 a	9.3 b	6.4 c	19.8 a	6.8 c	4.7 c
NH_4_^+^–N, mg kg^−1^	4.5 a	1.8 b	1.6 b	5.6 a	2.7 b	1.6 b
NO_3_^−^–N, mg kg^−1^	13.8 a	7.5 b	4.8 c	14.1 a	4.1 c	3.0 c
Available P, mg kg^−1^	26.8 a	26.0 a	25.3 ab	22.5 bc	22.1 cd	19.1 d
Exchangeable Ca, cmol_+_ kg^−1^	5.00 a	4.68 a	3.77 b	4.91 a	4.73 a	4.61 ab
Exchangeable Mg, cmol_+_ kg^−1^	0.43 bcd	0.49 ab	0.36 d	0.55 a	0.45 bc	0.38 cd
Exchangeable K, cmol_+_ kg^−1^	0.46 b	0.47 b	0.45 b	0.42 b	0.58 a	0.45 b
Exchangeable Na, cmol_+_ kg^−1^	0.067 ab	0.068 ab	0.032 c	0.052 b	0.074 a	0.078 a
Exchangeable Al, cmol_+_ kg^−1^	0.074 ab	0.056 b	0.061 b	0.046 b	0.094 a	0.066 b
Available S, mg kg^−1^	33.9 ab	26.8 cd	36.0 a	24.2 d	31.0 bc	27.4 cd

C: canola; BW: bread wheat; DW: durum wheat; B: bean. Different letters in the same row indicate differences between crop rotations as an average of the four levels of residue incorporation according to Tukey’s test (*p* < 0.05).

**Table 6 plants-14-00764-t006:** Soil chemical properties as affected by four levels of residue incorporation as average of six crop rotations.

Soil Properties	Level of Residue Incorporation (%)			
	0	50	100	200
pH	6.04 a	6.02 a	6.05 a	6.07 a
OM, g kg^−1^	10.2 a	10.5 a	10.1 a	10.1 a
Available N, mg kg^−1^	13.3 a	13.9 a	12.9 a	13.8 a
NH_4_^+^–N, mg kg^−1^	3.7 a	2.9 a	3.0 a	3.9 a
NO_3_^−^–N, mg kg^−1^	9.6 a	10.9 a	9.9 a	9.9 a
Available P, mg kg^−1^	25.5 a	26.2 a	24.5 a	24.4 a
Exchangeable Ca, cmol_+_ kg^−1^	4.25 a	4.47 a	4.65 a	4.99 a
Exchangeable Mg, cmol_+_ kg^−1^	0.43 b	0.44 b	0.45 ab	0.51 a
Exchangeable K, cmol_+_ kg^−1^	0.41 c	0.40 c	0.45 b	0.53 a
Exchangeable Na, cmol_+_ kg^−1^	0.05 a	0.06 a	0.05 a	0.05 a
Exchangeable Al, cmol_+_ kg^−1^	0.07 a	0.06 a	0.06 a	0.05 a
Available S, mg kg^−1^	31.1 a	30.3 a	29.8 a	29.8 a

Different letters in the same row indicate differences between crop rotations as an average of the four levels of residue incorporation according to Tukey’s test (*p* < 0.05).

**Table 7 plants-14-00764-t007:** Correlation matrix of different soil chemical properties at the end of the eight years of crop rotations.

	OM	N	P	Ca	Mg	K	Na	Al	S	N-NO_3_^−^	N-NH_4_^+^
pH	−0.43 ***	−0.11	−0.39 ***	0.64 ***	0.53 ***	0.13	0.17	−0.82 ***	−0.77 ***	−0.11	−0.06
OM		0.15	0.22 *	0.16	0.11	0.30 ***	0.27 ***	0.41 ***	0.21 *	0.17	0.01
N			0.17	0.18	0.30 **	−0.04	−0.06	−0.08	−0.09	0.96 ***	0.75 ***
P				−0.14	−0.19	−0.25 *	−0.24 *	0.31 **	0.20	0.21 *	−0.01
Ca					0.80 ***	0.46 ***	0.31 **	−0.52 ***	−0.71 ***	0.21 *	0.06
Mg						0.56 ***	0.17	−0.51 ***	−0.74 ***	0.33 ***	0.13
K							0.13	−0.07	−0.31 **	−0.06	0.01
Na								−0.03	−0.18	−0.06	−0.04
Al									0.63 ***	−0.10	−0.02
S										−0.10	−0.07
NO_3_^−^-N											0.54 ***

* Significant at *p* ≤ 0.05; ** significant at *p* ≤ 0.01; *** significant at *p* ≤ 0.001. *n* = 96.

## Data Availability

Data supporting reported results were generated during the study.

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
