# Peer review of "Chemical Properties of a Volcanic Soil Are Influenced by Eight Years of Crop Rotations with Different Levels of Residue Incorporation"

_plants, 2025, doi:10.3390/plants14050764_

Round 1

Reviewer 1 Report

Comments and Suggestions for Authors

The purpose of the study was to evaluate the effect of four cycles of six biannual rotations (canola-bread wheat, bean-bread wheat, canola-durum wheat, canola-durum wheat, canola-corn and bean-corn) and four levels of incorporation of residues for each crop (0%, 50%, 100% and 200%) on the chemical properties of a volcanic soil from central-southern Chile at the end of the evaluation period. According to authors the application of crop rotations with incorporation of residues is a conservationist practice that should improve in Latin America. It resulted from the theory that the incorporation of residues in increasing doses in crop rotations, gives a time window between harvesting and sowing, with the consequent improvement in soils chemical properties.

Although the problem of the positive effect of crop rotation and residues incorporation on soil quality is not new in the agricultural practice it is very important for soils in Latin America.

The study was correctly done, however the text needs some corrections. In the Abstract the study background should be replaced by the study aim and the chapters’ names should be removed.

The Introduction clearly presents the background of the study and uses appropriate literature. The research design is correct and methods used are correctly described. The description and interpretation of the obtained results is clear, however Table 3 looks very complicated. Conclusion cooperates with obtained results. The word Conclusion should be separated as an additional chapter title.

Author Response

The purpose of the study was to evaluate the effect of four cycles of six biannual rotations (canola-bread wheat, bean-bread wheat, canola-durum wheat, canola-durum wheat, canola-corn and bean-corn) and four levels of incorporation of residues for each crop (0%, 50%, 100% and 200%) on the chemical properties of a volcanic soil from central-southern Chile at the end of the evaluation period. According to authors the application of crop rotations with incorporation of residues is a conservationist practice that should improve in Latin America. It resulted from the theory that the incorporation of residues in increasing doses in crop rotations, gives a time window between harvesting and sowing, with the consequent improvement in soils chemical properties.

Although the problem of the positive effect of crop rotation and residues incorporation on soil quality is not new in the agricultural practice it is very important for soils in Latin America.

The study was correctly done, however the text needs some corrections. In the Abstract the study background should be replaced by the study aim and the chapters’ names should be removed.

In according with the reviewer 1 the Abstract was corrected including the aim of the study and the chapters names was removed.

The Introduction clearly presents the background of the study and uses appropriate literature. The research design is correct and methods used are correctly described. The description and interpretation of the obtained results is clear, however Table 3 looks very complicated. Conclusion cooperates with obtained results. The word Conclusion should be separated as an additional chapter title.

The conclusion was separated as an additional chapter title. However the Table 3 needs keep the structure because allow know all the crop rotations sequences and their grain yields, which too are used in the discussion for evaluate the climate effect during the last season on both durum wheat and bread wheat.

The changes were highlighted with red color.

Reviewer 2 Report

Comments and Suggestions for Authors

Interesting topic. Work written relatively correctly. Only 7 small remarks were marked. The main flaw of the work is the lack of a Conclusions chapter. This needs to be supplemented.

Author Response

All the corrections indicated for the Reviewer 2 were incorporated within the text and the changes were highlighted with red color. The rules for references were consulted when the manuscript was prepared for the first send.

Reviewer 3 Report

Comments and Suggestions for Authors

Your article is very interesting and provides valuable information. Please clarify the following methodological aspects:

Why was lime applied only in the areas of the rotation with canola?. If a difference was detected between the pH values of the experimental units, explain it in terms of homogeneity. Pag 4, 121-122

The decrease in soil pH associated with rotations that include canola can also be explained by the low initial soil pH values, and the added lime did not solve totally the problem pag 6, 171, 172, pag 12, 289, 290, 291.

Regarding the risk of there being or not a confounding effect on the results due to the application of lime only in a portion of the land, mention how the block design influenced the reduction of the effect of spatial variation in such a large experiment (1.34 ha).

Author Response

our article is very interesting and provides valuable information. Please clarify the following methodological aspects:

Why was lime applied only in the areas of the rotation with canola?. If a difference was detected between the pH values of the experimental units, explain it in terms of homogeneity. Pag 4, 121-122.

The paragraph was corrected because the lime was applied to all the experiment, previous to the sowing of all the crops in the first rotation cycle. The changes were highlighted in red color.

The decrease in soil pH associated with rotations that include canola can also be explained by the low initial soil pH values, and the added lime did not solve totally the problem pag 6, 171, 172, pag 12, 289, 290, 291.

Sorry, but there was an mistake in the explanation of the lime. The lime was applied to all the experiment previous to sowing of all the first crop cycle. The changes were highlighted with red color.

Regarding the risk of there being or not a confounding effect on the results due to the application of lime only in a portion of the land, mention how the block design influenced the reduction of the effect of spatial variation in such a large experiment (1.34 ha). 

I apologize for the error in the Material and Methods. The lime was applied to all the experiment (1.34 ha) previous the start the first cycle of crop rotation. The changes were highlighted with red color.

Round 2

Reviewer 2 Report

Comments and Suggestions for Authors

Work written relatively correctly. Almost all my comments were taken into account. The editing of the references was not corrected. So this one small comment remains. The Conclusion chapter has been nicely completed.
